# Adaptive dynamic programming-based optimal heading control for state constrained unmanned sailboat

1st Shitong Zhang
*School of Mechanical Engineering*
*Yanshan University*
Qinhuangdao, China
bben@stumail.ysu.edu.cn

2st Yifei Xu
*School of Mechanical Engineering*
*Yanshan University*
Qinhuangdao, China
xyf@ysu.edu.cn

3st Yingjie Deng*
*School of Mechanical Engineering*
*Yanshan University*
Qinhuangdao, China
dyj@ysu.edu.cn

4st Sheng Xu
*Shenzhen Institute of Advanced Technology*
*Chinese Academy of Sciences*
Shenzhen, China

*Abstract*—This paper proposes a new state-constrained adaptive optimal control strategy for unmanned sailboat heading angle tracking considering the motion constraints. To improve the tracking accuracy, a combination of backstepping and adaptive dynamic programming (ADP) is employed. Thus, the issue of virtual control rate derivatives in conventional backstepping control is resolved with satisfactory precision. Firstly, the motion constraints are considered by using the Barrier Lyapunov function (BLF), and the neural networks (NNs) is employed to approximate the model uncertainties and disturbances. Secondly, an adaptive backstepping feedforward controller is proposed, transforming the sailboat's affine nonlinear system tracking problem into a regulation problem. Thirdly, according to the ADP theory, critic NNs are constructed to approximate the analytical solution of the Hamilton-Jacobi-Bellman (HJB) equation, and the optimal feedback control is obtained by online learning. Finally, simulation results demonstrate the effectiveness and optimality of the proposed controller.

*Index Terms*—Optimal control, Adaptive dynamic programming (ADP), barrier Lyapunov function (BLF), Neural networks (NNs), Unmanned sailboat

## I. INTRODUCTION

In the past few decades, surface and underwater intelligent vehicles have played important roles in sea patrols, resource exploration, and rescue. However, due to the consumption of fuel, electricity, and other energy sources, unmanned ships and submarines require a large amount of power support to complete long-distance tasks, which also leads to huge cost problems in [1]–[6]. Since unmanned sailboats can use sails as power, have low costs, and can transmit data in real-time through their sensors, research on unmanned sailboats has gradually entered the interest of scholars. However, due to the

This work is partially supported by the Natural Science Foundation of China (No.52101375), the Hebei Province Natural Science Fund (No.E2022203088, E2024203179), the Innovation Capacity Enhancement Program of Hebei Province (No.24461901D), the Joint Funds of the National Natural Science Foundation of China (No.U20A20332), and the Key Research and Development Project of Hebei Province (No.21351802D).

complex marine environment, the disturbances of wind and waves can easily generate obstructive forces on the sails and keel, making the sails control difficult. Therefore, the authors of [7] proposed a path tracking method for unmanned sailboats that combines logic virtual ship (LVS) guidance law and dynamic event-triggered control. In [8], a path tracking control scheme is proposed for unmanned sailboats that combined backstepping and dynamic surface control technology. In order to reduce the sideslip angle error during sailing, the authors of [9] proposed double finite-time observers-based line-of-sight guidance (DFLOS) and adaptive finite-time control (DFLOS-AFC) strategies. However, they did not consider the issue of state constraints. When subjected to significant interference, to return the sailboat to the reference heading, a larger rudder angle is required, which will damage the actuator as it cannot accept significant deflection in a short period. Therefore, while ensuring the accuracy of sailboat heading control, it is also necessary to ensure that the turning speed of the boat heading satisfies the prescribed constraints.

In order to solve the problem of unmanned sailboat heading angle tracking accuracy, many scholars have conducted research on it. In [10], a control strategy of adaptive echo state networks and backstepping is proposed to control the steering angle of the rudder. The authors of [11] designed a nonlinear heading controller of velocity vector direction to track the reference heading angle. In [12], the $L_1$ adaptive control theory is proposed to complete heading control and ensure the stability of the required heading angle. However, most papers focused on the heading control of sailboats using backstepping or sign functions, which results in low tracking accuracy and cannot achieve optimal control effects.

The key problem of the nonlinear optimal control is that the analytical solution of the Hamilton-Jacobi-Bellman (HJB) equation is difficult to solve. In order to obtain the analytical solution, the authors of [13] proposed the ADP theory, which

approximates the solution of the equation online. However, the problem of slow approximation speed and multiple iterations has led to an explosion in computational complexity. Recently, with the rapid development of NNs, their approximation performance and speed for unknown nonlinear functions have become increasingly excellent. Therefore, the research combining ADP and NNs has solved the above problems. In [14], a new event-triggered optimal trajectory tracking control method based on goal representation heuristic dynamic programming (GrHDP) for underactuated ships is proposed. In [15], they proposed a model free dual heuristic dynamic programming (DHP) method for unmanned aerial vehicle attitude control. The authors of [16] applied ADP theory to the path planning problem of mobile robots. However, to the best of our knowledge, there is limited research on applying ADP theory to the heading control of unmanned sailboats.

Taking inspiration from the above analysis, we introduce the LBF function to solve the state constraint problem. In the control design of heading tracking, while introducing the backstepping method, the ADP theory is also introduced to ensure the optimal tracking accuracy. We constructed an evaluation NN to approximate the analytical solution of the HJB equation and obtained the optimal feedback control through online learning. The main contributions are concluded below:

- Compared with previous papers of [7]–[9], we proposed a LBF-based method to overcome the issues of the motion constraints caused by the turning rate limitation of unmanned sailboats.
- By using the ADP optimal feedback control strategy, our designed control method achieves optimal tracking accuracy in heading control compared to [9]–[12].
- According to the construction of the critic network, a training speed acceleration method is developed using online learning methods.

The remaining of this paper is organized as follows. Section II elaborates on the heading angle control model of unmanned sailboats and lemma. Section III designs the backstepping feedforward controller and the optimal feedback controller. Section IV provides simulation verification of the proposed strategy superiority. Section V gives some conclusions.

## II. MATHEMATICAL MODEL AND PRELIMINARIES

The sailboat is divided into four parts: the sail, rudder, keel, and hull. Combined with the theory of gas fluid dynamics, force analysis is conducted on each part, ignoring the undulating and pitching motion of the sailboat to establish a 3-DOF mathematical model of sailboat motion. The sailboat model considering external interference and control input rudder angle in this paper is

$$\begin{cases} \dot{\psi} = r \\ \dot{r} = f_r(\psi, r, \delta_s, u, v, \tau_{wr}) + g_r \tau_r \end{cases} \tag{1}$$

where $\psi \in \mathbb{R}^M$ and $r \in \mathbb{R}^M$ are system state variables, respectively; $u \in \mathbb{R}^M$ is the forward velocity; $v \in \mathbb{R}^M$ is the lateral velocity; $\delta_s \in \mathbb{R}^M$ is the sail angle; $\tau_{wr}$ is the external

disturbance; $f_r(\cdot)$ is the nonlinear function of the unknown model of a sailboat; $g_r$ is the unknown control gain; $\tau_r$ is the control input of the system.

**Assumption 1 [17]:** Due to the fact that unmanned sailboats navigate in limited space, there exists a normal number that satisfies the heading angle and heading angular speed to be less than or equal to this normal number. That is, $\psi \leq k_\psi$ and $r \leq k_r$.

**Lemma 1 [10]:** Due to the outstanding ability of NNs in function approximation, they are often used to approximate nonlinear functions. Therefore, NNs can be used to approximate unknown functions as follows:

$$F(x) = W^{\mathrm{T}}\sigma(x) + \varepsilon(x) \tag{2}$$

where $W = (w_1, w_2, ..., w_n)^{\mathrm{T}}$ denotes the desired weight of the NNs and $\varepsilon \leq \bar{\varepsilon}$ denotes the approximating error.

## III. CONTROL DESIGN

### A. Feedforward controller design

In this section, the backstepping method based on an adaptive NNs framework is adopted to transform system (1) into an affine nonlinear system.

According to the system (1), define the error system function as

$$\begin{cases} \psi_e = \psi - \psi_d \\ r_e = r - r_d \end{cases} \tag{3}$$

where $\psi_d$ and $r_d$ are the reference heading angle and yaw speed, respectively. Differentiating $\psi_e$ along with (1), it has:

$$\dot{\psi}_e = \dot{\psi} - \dot{\psi}_d = r - \dot{\psi}_d = (r_e + r_d) - \dot{\psi}_d \tag{4}$$

where $r_d = r_d^\alpha + r_d^*$ is the virtual control input of yaw speed. $r_d^\alpha$ denotes the feedforward virtual yaw speed input and $r_d^*$ denotes the feedback virtual yaw speed input. Therefore, we can get

$$\dot{\psi}_e = r_e + r_d^\alpha + r_d^* - \dot{\psi}_d \tag{5}$$

To construct the desired feedforward yaw speed virtual control input, consider the BLF as

$$V_1 = \frac{1}{2} \log\left(\frac{k_\psi^2}{k_\psi^2 - \psi_e^2}\right) \tag{6}$$

where $k_\psi$ is a positive value of the state constraint. Calculate the derivate of $V_1$, we have

$$\dot{V}_1 = \frac{\psi_e}{k_\psi^2 - \psi_e^2}(r_e + r_d^\alpha + r_d^* - \dot{\psi}_d) \tag{7}$$

Therefore, the feedforward virtual yaw speed input $r_d^\alpha$ can be designed as

$$r_d^\alpha = -(k_\psi^2 - \psi_e^2)k_1\psi_e + \dot{\psi}_d \tag{8}$$

where $k_1 > 0$ is a tuning parameters. Substituting (8) into (7), we have

$$\dot{V}_1 = -k_1\psi_e^2 + \frac{\psi_e}{k_\psi^2 - \psi_e^2}(r_e + r_d^*) \tag{9}$$

Taking the derivative of the second equation of (3) yields

$$\dot{r}_e = f_r(\cdot) + f_r(e_d) - f_r(e_d) + g_r\tau_r - \dot{r}_d \qquad (10)$$

where $e_d = [\psi_d, r_d]^\mathrm{T}$. The unknown model uncertainty function $f_r(\cdot)$ and $\dot{r}_d$ can be transferred by the following function

$$F_2(z_{2d}) = f_r(e_d) - \dot{r}_d \qquad (11)$$

where $z_{2d} = [e_d^\mathrm{T}, \psi_e, r_e]^\mathrm{T}$. According to the Lemma 1, the above (11) can be approximated by NNs as follows:

$$F_2(z_{2d}) = (\hat{W}_2^\mathrm{T} + \tilde{W}_2^\mathrm{T})\sigma_2(z_{2d}) + \varepsilon_2(z_{2d}) \qquad (12)$$

where $\tilde{W}_2^\mathrm{T} = W_2 - \hat{W}_2$ is the NNs approximate error and $\hat{W}_2$ is the estimation of optimal weight vector $W_2$. Through (11) and (12), $f_r(\cdot) - f_r(e_d)$ can be approximated as follows:

$$\begin{aligned}
f_r(\cdot) - f_r(e_d) =& F_2(z_2) - F_2(z_{2d}) \\
=& p(e) + \tilde{W}_2^\mathrm{T}[\sigma_2(z_2) - \sigma_2(z_{2d})] + \varepsilon_2(z_2) \\
& - \varepsilon_2(z_{2d})
\end{aligned} \qquad (13)$$

where $p(e) = \hat{W}_2^\mathrm{T}\sigma_2(z_2) - \hat{W}_2^\mathrm{T}\sigma_2(z_{2d})$, $e = [\psi_e, r_e]^\mathrm{T}$ and $F_2(z_2) = f_r(\cdot) - \dot{r}_d$ is a function of $r_d$ and $\dot{r}_d$. The input $z_2$ is chosen as $(e_d^\mathrm{T}, \psi, r, \delta_s, u, v)^\mathrm{T}$. According to (12) and (13), (10) can be written as

$$\begin{aligned}
\dot{r}_e =& p(e) + \hat{W}_2^\mathrm{T}\sigma_2(z_{2d}) + \tilde{W}_2^\mathrm{T}\sigma_2(z_2) + \varepsilon_2(z_2) \\
& + g_r\tau_r^\alpha + g_r\tau_r^*
\end{aligned} \qquad (14)$$

To construct the feedforward virtual control input $\tau_r^\alpha$, consider the BLF as

$$V_2 = V_1 + \frac{1}{2}\log\left(\frac{k_r^2}{k_r^2 - r_e^2}\right) + \frac{1}{2}\tilde{W}_2^\mathrm{T}\tilde{W}_2 \qquad (15)$$

where $k_r$ is a positive value of the motion constraint. Calculate the derivate of $V_2$, we have

$$\begin{aligned}
\dot{V}_2 =& -k_1\psi_e^2 + \frac{\psi_e}{k_\psi^2 - \psi_e^2}(r_e + r_d^*) + \frac{r_e}{k_r^2 - r_e^2}\Big(p(e) \\
& + \hat{W}_2^\mathrm{T}\sigma_2(z_{2d}) + \tilde{W}_2^\mathrm{T}\sigma_2(z_2) + \varepsilon_2(z_2) + g_r\tau_r^\alpha \\
& + g_r\tau_r^*\Big) - \tilde{W}_2^\mathrm{T}\dot{\hat{W}}_2
\end{aligned} \qquad (16)$$

According to the Young's inequality, we can get that

$$\frac{r_e}{k_r^2 - r_e^2}\varepsilon_2(z_2) \le \frac{r_e}{k_r^2 - r_e^2}\bar{\varepsilon}_2 \le \frac{1}{2}\frac{r_e^2}{(k_r^2 - r_e^2)^2} + \frac{1}{2}\bar{\varepsilon}_2^2 \quad (17)$$

Substituting (17) into (16), we have

$$\begin{aligned}
\dot{V}_2 =& -k_1\psi_e^2 + \frac{1}{2}\bar{\varepsilon}_2^2 - \tilde{W}_2^\mathrm{T}\dot{\hat{W}}_2 + \frac{\psi_e r_e}{k_\psi^2 - \psi_e^2} + \frac{\psi_e r_d^*}{k_\psi^2 - \psi_e^2} \\
& + \frac{r_e}{k_r^2 - r_e^2}\Big(p(e) + \hat{W}_2^\mathrm{T}\sigma_2(z_{2d}) + \tilde{W}_2^\mathrm{T}\sigma_2(z_2) \\
& + \varepsilon_2(z_2) + g_r\tau_r^\alpha + g_r\tau_r^*\Big)
\end{aligned} \qquad (18)$$

Therefore, the feedforward control input $\tau_r^\alpha$ can be designed as

$$\tau_r^\alpha = -\frac{1}{g_r}\left[(k_r^2 - r_e^2)k_2 r_e + \frac{(k_r^2 - r_e^2)\psi_e}{k_\psi^2 - \psi_e^2} + \hat{W}_2^\mathrm{T}\sigma_2(z_{2d})\right] \qquad (19)$$

where $k_2 > 0$ is a tuning parameters. The NNs weight vector adaptation law $\hat{W}_2$ can be designed as

$$\dot{\hat{W}}_2 = \frac{r_e}{k_r^2 - r_e^2}\sigma_2(z_2) - \beta_2\hat{W}_2 \qquad (20)$$

where $\beta_2 > 0$ is also a tuning parameters. Substituting (19) and (20) into (18), we have

$$\begin{aligned}
\dot{V}_2 \le& -k_1\psi_e^2 - k_2 r_e^2 + \frac{1}{2}\bar{\varepsilon}_2^2 + \beta_2\tilde{W}_2^\mathrm{T}\hat{W}_2 + \frac{p(e)r_e}{k_r^2 - r_e^2} \\
& + \frac{r_e g_r}{k_r^2 - r_e^2}\tau_r^* + \frac{\psi_e r_d^*}{k_\psi^2 - \psi_e^2}
\end{aligned} \qquad (21)$$

According to the Young's inequality, we can get that

$$\begin{aligned}
\tilde{W}_2^\mathrm{T}\hat{W}_2 &= \tilde{W}_2^\mathrm{T}(W_2 - \tilde{W}_2) = \tilde{W}_2^\mathrm{T}W_2 - \tilde{W}_2^\mathrm{T}\tilde{W}_2 \\
&\le \frac{1}{2}\tilde{W}_2^\mathrm{T}\tilde{W}_2 + \frac{1}{2}W_2^\mathrm{T}W_2 - \tilde{W}_2^\mathrm{T}\tilde{W}_2 \\
&= \frac{1}{2}W_2^\mathrm{T}W_2 - \frac{1}{2}\tilde{W}_2^\mathrm{T}\tilde{W}_2
\end{aligned} \qquad (22)$$

Therefore, the above (22) can be written as

$$\begin{aligned}
\dot{V}_2 \le& -\underline{k}\|E\|^2 + \frac{1}{2}\bar{\varepsilon}_2^2 + \frac{1}{2}\beta_2 W_2^\mathrm{T}W_2 - \frac{1}{2}\tilde{W}_2^\mathrm{T}\tilde{W}_2 \\
& + \frac{p(e)r_e}{k_r^2 - r_e^2} + \frac{r_e g_r}{k_r^2 - r_e^2}\tau_r^* + \frac{\psi_e r_d^*}{k_\psi^2 - \psi_e^2}
\end{aligned} \qquad (23)$$

where $E = [\psi_e, r_e]\mathrm{T}$, $\underline{k} = \min(k_1, k_2)$.

In previous research, the feedforward controller $\tau_r^\alpha$ was calculated based on the derivative of the virtual controller $\dot{r}_d^\alpha$. However, in practical applications, it is not easy to get analytical solutions for $\dot{r}_d^\alpha$. Our proposed control method denotes $r_d = r_d^\alpha + r_d^*$, which is determined by both $r_d^\alpha$ and $r_d^*$ and they are obtained through NNs weights and the system's state. Therefore, we approximate the derivative of the virtual controller using NNs in (11). With this approach, the feedforward controller $r_d^\alpha$ and $\tau_r^\alpha$ can be obtained directly from the current system without the need for the derivative of the virtual controller used in previous studies. Consequently, compared to previous work, the method we propose is more feasible to implement in practical applications.

Rewriting (23) as follow:

$$\begin{aligned}
\dot{V}_2 \le& -\underline{k}\|E\|^2 + \frac{1}{2}\bar{\varepsilon}_2^2 + \frac{1}{2}\beta_2 W_2^\mathrm{T}W_2 - \frac{1}{2}\tilde{W}_2^\mathrm{T}\tilde{W}_2 \\
& + \frac{E^\mathrm{T}}{\tilde{E}^\mathrm{T}}\left(\begin{bmatrix} 0 \\ p(e) \end{bmatrix} + \begin{bmatrix} 1 & 0 \\ 0 & g_r \end{bmatrix}\begin{bmatrix} r_d^* \\ \tau_r^* \end{bmatrix}\right)
\end{aligned} \qquad (24)$$

where $\tilde{E} = [k_\psi^2 - \psi_e^2, k_r^2 - r_e^2]^\mathrm{T}$. The feedforward controller is expressed as $U^\alpha = [r_d^\alpha, \tau_r^\alpha]$. The feedback optimal controller $U^* = [r_d^*, \tau_r^*]$ will be designed in the subsection. Therefore, they constitute the controller of the entire system.

## B. Feedback optimal controller design

According to (24), the design of an individual feedforward controller for $U^\alpha$ cannot guarantee the stability of the entire closed-loop system. Therefore, to ensure the stability of the last term in (24), a feedback optimal controller is designed based on ADP theory. With this design, not only the tracking ability of the system can be optimized, but also the system's stability can be ensured.

The last term in (24) can be written as:

$$\dot{E} = \begin{bmatrix} 0 \\ p(e) \end{bmatrix} + \begin{bmatrix} 1 & 0 \\ 0 & g_r \end{bmatrix} U^* \tag{25}$$

Further, it can be obtained that

$$\dot{E} = P(E) + G\hat{U} \tag{26}$$

where $E = [\psi_e, r_e]^T$ are the heading angle error and yaw speed error, $P(E) = [0, p(e)]^T$, $G = diag[1, g_r]^T$.

According to the ADP theory, the performance index function can be define as

$$J(E) = \int_t^\infty E^T Q E + \hat{U}^T R \hat{U} \mathrm{d}\tau \tag{27}$$

where $Q \in \mathbb{R}^{2\times 2}$ and $R \in \mathbb{R}^{2\times 2}$ are positive definite matrices.

The Hamiltonian function of the performance index function can be defined as

$$H(E, \hat{U}, \nabla J(E)) = E^T Q E + \hat{U}^T R \hat{U} + \nabla J(E)^T \Big( P(E) + G\hat{U} \Big) \tag{28}$$

where $\nabla J(E) = \frac{\partial J(E)}{\partial E}$ denotes the derivative of $J(E)$ with regard to $E$. In order to solve the HJB equation, the feedback optimal control $U^*$ can be designed as

$$U^*(E) = -\frac{1}{2}R^{-1}G^T \nabla J^*(E) \tag{29}$$

From (28), the optimal performance index function $J^*(E)$ can be obtained by

$$\min_{\hat{U}(E)} H\Big(E, \hat{U}, \nabla J^*(E)\Big) = 0 \tag{30}$$

Substituting the above (30) into (28), the HJB equation can be rewritten as follow:

$$E^T Q E + \big(\nabla J^*(E)\big)^T P(E) - \frac{1}{4}\Big(\big(\nabla J^*(E)\big)^T \\ G R^{-1} G^T \nabla J^*(E)\Big) = 0 \tag{31}$$

It is obvious that the above equation is a nonlinear partial differential equation, so it is difficult to obtain its analytical solution. Therefore, to address this issue, the ADP theory is adopted. By constructing a single-layer NN to approximate the following optimal performance index function as

$$J^*(E) = W_c^T \sigma(E) + \varepsilon_c(E) \tag{32}$$

where $W_c$ denotes the optimal weight vector of critic NNs, $\sigma(\cdot)$ is the activation function, $\varepsilon_c(E)$ is the critic NNs approximation error.

The gradient of the optimal performance index function $J^*(E)$ with regard to $E$ can be defined as

$$\nabla J^*(E) = \big(\nabla \sigma(E)\big)^T W_c + \nabla \varepsilon_c(E) \tag{33}$$

From (33) and (32), (29) can be rewritten as

$$U^*(E) = -\frac{1}{2}R^{-1}G^T\big(\nabla \sigma(E)\big)^T W_c - \frac{1}{2}R^{-1}G^T \nabla \varepsilon_c(E) \tag{34}$$

Therefore, the HJB equation can be further designed as

$$H(E, U^*, W_c) = E^T Q E + W_c^T \nabla \sigma(E) P(E) + \varepsilon_{HJB} \\ - \frac{1}{4}\Big(W_c^T \nabla \sigma(E) G R^{-1} G^T \big(\nabla \sigma(E)\big)^T W_c\Big) \\ = 0 \tag{35}$$

where $\varepsilon_{HJB}$ is the error.

By using NNs to estimate the desired weights of performance index function as follow:

$$\hat{J}(E) = \hat{W}_c^T \sigma(E) \tag{36}$$

where $\hat{W}_c$ and $\hat{J}(E)$ are estimations of $W_c$ and $J(E)$, respectively. Let weight estimation error as $\tilde{W}_c = W_c - \hat{W}_c$, the estimate of optimal control $U^*$ can be designed as

$$\bar{U}(E) = -\frac{1}{2}R^{-1}G^T\big(\nabla \sigma(E)\big)^T \hat{W}_c \tag{37}$$

Then, the HJB equation can be approximated as

$$H(E, \bar{U}, W_c) = E^T Q E + W_c^T \nabla \sigma(E) P(E) \\ - \frac{1}{4}\Big(\hat{W}_c^T \nabla \sigma(E) G R^{-1} G^T \big(\nabla \sigma(E)\big)^T \hat{W}_c\Big) \\ = e_c \tag{38}$$

The objective error function of critic NNs is defined as

$$E_c = \frac{1}{2}e_c^2 \tag{39}$$

From the [18], we can design a appropriate critic NNs updating law, which can guarantee that $\hat{W}_c$ converges to $W_c$ and also minimize the objective error function (39).

$$\dot{\hat{W}}_c = -k_c\frac{\Gamma}{(1+\Gamma^T\Gamma)^2}e_c + \frac{k_c}{2}\Delta\nabla\sigma(E)G\nabla V(E) \\ + k_c\left[\frac{1}{4}\frac{\Gamma}{(1+\Gamma^T\Gamma)^2}\hat{W}_c^T\nabla\sigma(E)G\big(\nabla\sigma(E)\big)^T\hat{W}_c\right] \\ + k_c\left[K_1\zeta^T\hat{W}_c - K_2\hat{W}_c\right] \tag{40}$$

where $k_c > 0$ is the tuning parameter, $\Gamma = \nabla\sigma(E)(P(E) + G\bar{U})$, $\zeta = \frac{\Gamma}{1+\Gamma^T\Gamma}$, $K_1$ and $K_2$ are the tuning parameter. $\Delta$ is designed as

$$\Delta = \begin{cases} 0, \big(\nabla V(E)\big)^T\big(P(E) + G\bar{U}\big) < 0 \\ 1, \text{else} \end{cases} \tag{41}$$

where $V(E)$ is a Lyapunov function. From this, we can obtain

$$
\begin{aligned}
\dot{V}(E) &= \left(\nabla V(E)\right)^{\mathrm{T}} \dot{E} = \left(\nabla V(E)\right)^{\mathrm{T}} \left(P(E) + GU^*\right) \\
&= -\left(\nabla V(E)\right)^{\mathrm{T}} S \nabla V(E) \leq 0
\end{aligned}
\tag{42}
$$

where $S$ is a positive definite matrix. Specifically, $V(E)$ is a function of the state variable $E$ and can be chosen appropriately, for example, $V(E) = E^{\mathrm{T}} E$.

**Remark 1:** The weight $\hat{W}_c$ update process consists of the following four components: The first component employs gradient descent for design. The second component ensures the boundedness of the weights. The third and fourth components guarantee the stability of the weights. Through this design, the proposed control strategy achieves a higher tracking accuracy while ensuring the rapid and stable update of the neural network weights.

## IV. SIMULATION

The model parameters of the unmanned sailboat are selected from [10]. In order to facilitate simulation analysis without losing generality, the reference heading is set as $\psi_d = sin(\mathrm{t})$. Select control parameters as $k_\psi = 1.2$, $k_r = 1.5$, $k_1 = 3$, $k_2 = 6$, $k_c = 3.8$, $\beta_2 = 4$, $K_1 = 0.0001\mathrm{I}$, $K_2 = 0.00001\mathrm{I}$, $Q = \mathrm{I}$, $R = 0.25\mathrm{I}$. The time step is set as 0.05. The initial states are defined as $\psi(0) = 0.05$ and $r(0) = 0$. The activation function of the critic network is chosen as $\sigma(E) = [e_1, e_1^2, e_2, e_2^2, e_1 e_2]$, the network weights are selected randomly in [0, 1]. Simulate real ocean and wind disturbances by using first-order Markov perturbations.

To verify the superiority of the proposed strategy, we will compare the "LSBG" strategy of [10]. Fig. 1 shows the real-time curve of heading angle tracking, and the results show that the designed optimal control method can track the reference signal with smaller errors and within state constraints. The heading angular velocity tracking and its state constraints are shown in Fig. 2. Fig. 3 and Fig. 4 illustrate the error curves of heading angle and heading angular velocity, indicating that the proposed strategy achieves better tracking performance than the "LSBG" strategy. Fig. 5 displays the control inputs for control input $\tau_r$ under "LSBG" strategy, backstepping feedforward control, optimal feedback control, and system control under the proposed strategy, respectively. Fig. 6 shows the update curve of the evaluation network weights, it is obvious that the speed of online learning has reached stability in a very short time. From the above analysis, the proposed strategy can not only ensure better tracking accuracy, but also avoid system state violations of constraints.

## V. CONCLUSION

In this paper, the optimal control method based on ADP is proposed for the tracking control of unmanned sailboats with heading turning constraints. The proposed LBF-based method solves the problem of state constraints. The feedforward backstepping controller and the feedback optimal controller were designed using the backstepping method and ADP theory, respectively. The learning ability of critic NNs has been

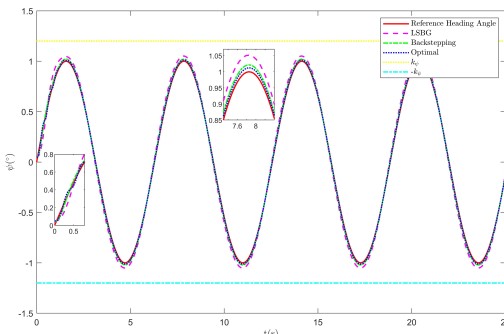

Fig. 1.  Comparison of heading angle tracking under different strategies.

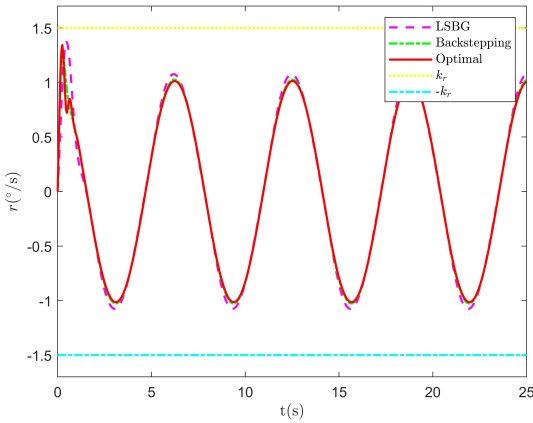

Fig. 2.  Comparison of heading angle speed tracking under different strategies.

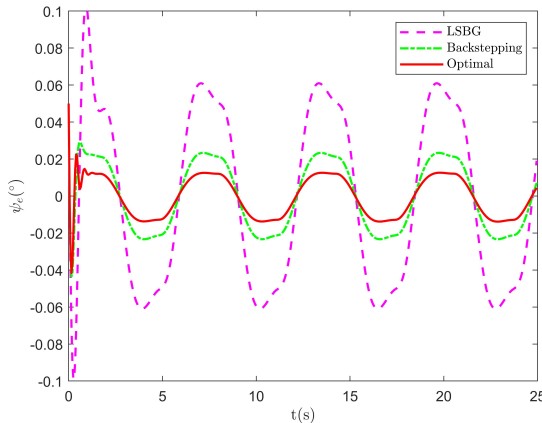

Fig. 3.  Comparison of heading angle error under different strategies.

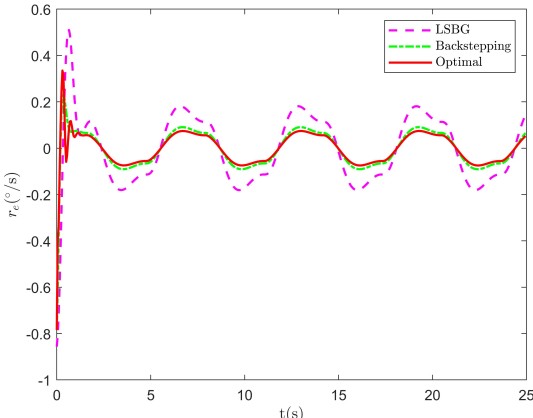

Fig. 4. Comparison of heading angle speed error under different strategies.

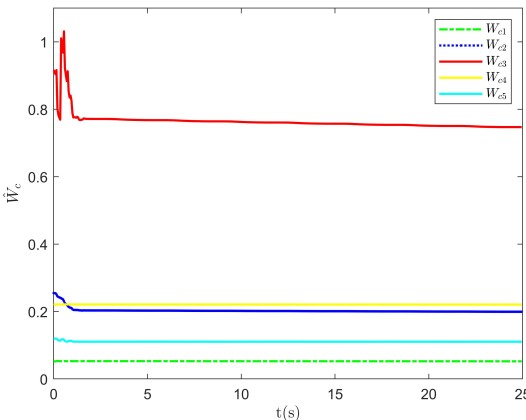

Fig. 6. Critic ADP weight update curve.

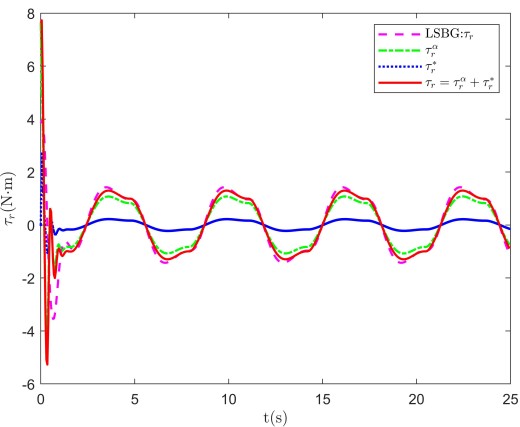

Fig. 5. Control input $\tau_r$ under "LSBG" strategy, feedforward control input $\tau_r^\alpha$, feedback control input $\tau_r^*$ and optimal control input $\tau_r$.

accelerated through online learning strategies. Finally, the simulation verified the optimality of the proposed strategy. In the future, we will apply this method to the path-tracking task of unmanned sailboats in pratice.

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
