# OpenReview forum: "Adaptive dynamic programming-based optimal heading control for state constrained unmanned sailboat"
_IEEE.org/ICIST/2024/Conference — IEEE ICIST 2024 Conference Submission_

### Official Review · Reviewer_QwjL · 2024-08-25
**Accept, some mistakes should be revised**

**Rating:** 7
**Confidence:** 3

**Review:**

1. The essential difference between the contributions and the existing works should be highlighted.
2. The performance of the designed strategy is insufficient. Detailed demonstrations should be supplied to the simulation results.

---

### Official Review · Reviewer_i86X · 2024-08-26
**The topic under consideration is interesting. This paper can be accepted after some modifications.**

**Rating:** 7
**Confidence:** 3

**Review:**

In this paper, a new state-constrained adaptive optimal control strategy is proposed for unmanned sailboat heading angle tracking considering the motion constraints. The topic under consideration is interesting. Detailed comments and suggestions are listed as follows.
1.	The English writing of the paper needs to be further polished, and some typos should be fixed, such as, “Substituting the above (8) into the (7), and it has…” and “Calculate the derivate of V1, and it has…”.
2.	The detailed process of controller design cannot be easily followed. Please give an important theorem about the feasibility of the proposed controller on Chapter III.
3.	The format of References should be standardized.

---

### Official Review · Reviewer_nNaL · 2024-08-31
**This paper proposes a new state-constrained adaptive optimal control strategy for unmanned sailboat heading angle tracking considering the motion constraints. The tracking accuracy is satisfactory.**

**Rating:** 7
**Confidence:** 5

**Review:**

1.Ensuring that the ADP algorithms converge to an optimal or near-optimal solution is challenging, especially in complex and high-dimensional environments, how to ensure convergence in this article
2.HJB equations are typically nonlinear partial differential equations. The nonlinearity complicates both analytical and numerical solutions, making it challenging to find precise solutions. How do the authors consider this problem?
3.The formula numbers mentioned in the paper should not have "the" in front, it is recommended to make modifications.
4.Please add a physical description on the vertical coordinate in Figures 6.

---

### Decision · Program_Chairs · 2024-09-06

Accept (Oral)